# Synthesis and Antiviral Activity of Camphene Derivatives against Different Types of Viruses

**DOI:** 10.3390/molecules26082235

**Published:** 2021-04-13

**Authors:** Anastasiya S. Sokolova, Valentina P. Putilova, Olga I. Yarovaya, Anastasiya V. Zybkina, Ekaterina D. Mordvinova, Anna V. Zaykovskaya, Dmitriy N. Shcherbakov, Iana R. Orshanskaya, Ekaterina O. Sinegubova, Iana L. Esaulkova, Sophia S. Borisevich, Nikolay I. Bormotov, Larisa N. Shishkina, Vladimir V. Zarubaev, Oleg V. Pyankov, Rinat A. Maksyutov, Nariman F. Salakhutdinov

**Affiliations:** 1N.N. Vorozhtsov Novosibirsk Institute of Organic Chemistry SB RAS, Lavrent’ev av., 9, 630090 Novosibirsk, Russia; valya1put1@gmail.com (V.P.P.); ooo@nioch.nsc.ru (O.I.Y.); mordvinova97@mail.ru (E.D.M.); anvar@nioch.nsc.ru (N.F.S.); 2State Research Center of Virology and Biotechnology VECTOR, Rospotrebnadzor, 630559 Novosibirsk, Russia; zybkina_av@vector.nsc.ru (A.V.Z.); zaykovskaya_av@vector.nsc.ru (A.V.Z.); dnshcherbakov@gmail.com (D.N.S.); bormotov_ni@vector.nsc.ru (N.I.B.); shish@vector.nsc.ru (L.N.S.); pyankov@vector.nsc.ru (O.V.P.); maksyutov_ra@vactor.nsc.ru (R.A.M.); 3Pasteur Institute of Epidemiology and Microbiology, 14 Mira str., 197101 St. Petersburg, Russia; deina89@mail.ru (I.R.O.); sinek489@gmail.com (E.O.S.); ianaesaulkova@gmail.com (I.L.E.); zarubaev@gmail.com (V.V.Z.); 4Laboratory of Chemical Physics, Ufa Institute of Chemistry Ufa Federal Research Center, 71 Pr. Oktyabrya, 450078 Ufa, Russia; monrel@yandex.ru

**Keywords:** camphen, antiviral agent, surface protein, pseudotype viruses, molecular docking

## Abstract

To date, the ‘one bug-one drug’ approach to antiviral drug development cannot effectively respond to the constant threat posed by an increasing diversity of viruses causing outbreaks of viral infections that turn out to be pathogenic for humans. Evidently, there is an urgent need for new strategies to develop efficient antiviral agents with broad-spectrum activities. In this paper, we identified camphene derivatives that showed broad antiviral activities in vitro against a panel of enveloped pathogenic viruses, including influenza virus A/PR/8/34 (H1N1), Ebola virus (EBOV), and the Hantaan virus. The lead-compound **2a**, with pyrrolidine cycle in its structure, displayed antiviral activity against influenza virus (IC_50_ = 45.3 µM), Ebola pseudotype viruses (IC_50_ = 0.12 µM), and authentic EBOV (IC_50_ = 18.3 µM), as well as against pseudoviruses with Hantaan virus Gn-Gc glycoprotein (IC_50_ = 9.1 µM). The results of antiviral activity studies using pseudotype viruses and molecular modeling suggest that surface proteins of the viruses required for the fusion process between viral and cellular membranes are the likely target of compound **2a**. The key structural fragments responsible for efficient binding are the bicyclic natural framework and the nitrogen atom. These data encourage us to conduct further investigations using bicyclic monoterpenoids as a scaffold for the rational design of membrane-fusion targeting inhibitors.

## 1. Introduction

In recent years, multiple outbreaks of infectious diseases have caused a serious threat to human health and the world economy. Due to tourism growth and trade globalization, the spread of infectious diseases is becoming increasingly rapid and extensive. New pathogens are emerging, with a striking example being the global pandemic of the new coronavirus infection of 2019 (COVID-19) caused by the SARS-CoV-2 virus, having started in Wuhan, Hubei province in China, and rapidly spreading around the world [1]. The agricultural expansion, deforestation, population growth, urbanization, close contact between humans and pathogen-infected animals led to an increasing pandemics outbreak of a zoonotic origin [2]. For example, monkeypox is a zoonotic, smallpox-like disease results from infection with Monkeypox virus (MPXV), a member of the *Orthopoxvirus* (OPV) genus, being transmitted to humans most commonly from infected rodents. Sporadic cases of human MPXV infection have been reported since 1970. However, there has been a recent increase in monkeypox cases, with 24,399 suspected monkeypox cases reported by 2018 and lethality accounting for up to 10% [3]. Such new or known infections that have recently increased in number are referred to as emerging and re-emerging infectious diseases (EIDs). EIDs include novel previously undescribed human diseases, new variants of previously known pathogens, or older pathogens that have emerged in new populations due to changes in human behavior or modifications to natural habitats. Examples of novel viruses include HIV, the Ebola virus, and SARS viruses. The influenza virus that caused the 2009 swine flu pandemic is an example of an existing virus that mutated and caused epidemics with considerable virulence. The lack of effective therapies for emerging and re-emerging virus infections represents an ongoing threat to public health. Given the increasing number of newly emerging viruses, the conventional “one-bug-one-drug” paradigm is insufficient, and developing broad-spectrum antiviral agents is essential to address the challenge of viral infections. 

A few antiviral agents with broad-spectrum characteristics are currently available. For example, RNA-dependent RNA polymerase inhibitor favipiravir exhibits broad-spectrum activity against RNA viruses, including arenavirus, bunyavirus, filovirus [4]. A nucleotide analog cidofovir and its oral bioavailable analog brincidofovir have been reported to demonstrate activity against DNA viruses, including herpes, polyomavirus, adenovirus, and pox viral families [5]. Ribavirin is a purine nucleoside analog with demonstrated efficacy against various DNA and RNA viral infections such as RSV (respiratory syncytial virus), hepatitis C, influenza A and B, parainfluenza viruses, and adenoviruses [6]. It is worth noting that the antiviral agents mentioned above have high toxicity and serious side effects. Thus, developing a new effective broad-spectrum antiviral agent is a top priority of medicinal chemistry.

Natural products have a wide range of pharmacophores and stereochemistry, and these properties are believed to contribute to the ability of natural products to display a wide range of biological activities [7]. Currently, natural products and their derivatives represent over one-third of all new molecular entities approved by the FDA (Food and Drug Administration) [8]. Recently our research group has conducted a large number of studies aimed at identifying the antiviral activity of derivatives synthesized from monoterpenoids such as (+)-camphor and (−)-borneol [9]. In particular, a number (−)-borneol based esters with N-containing heterocycles showed antiviral activity against different virus infections such as influenza virus [10], filoviruses [11,12], and orthopoxviruses including Variola virus (VARV), the causative agent of smallpox [13] (Figure 1). 

The ester group in compounds **Ia–d** can be unstable in vivo [14]. For example, it can be hydrolyzed by different enzymes, including varying cellular esterases present in various tissues and plasma [15]. To find antiviral agents with more stable functional groups, we synthesized a series of ethers with two crucial structural fragments: 1,7,7-trimethylbicyclo(2.2.1)heptan scaffold and N-containing saturated heterocyclic ring. To study the possibility for these agents to demonstrate a broad-spectrum antiviral activity, we evaluated the antiviral activity against RNA viruses such as influenza virus A/Puerto Rico/8/34 (H1N1), filoviruses, the Hantaan virus, and the DNA vaccinia virus.

## 2. Results and Discussion

### 2.1. Chemistry

All target compounds were synthesized in two steps. At the first step, the alkylation of 2-bromoethanol and 3-bromopropan-1-ol with (±)-camphene in the presence of heterogeneous catalysts montmorillonite clay K-10 led to the formation of bromides **1a–b**. The addition of alcohols to the olefin was followed by a Wagner–Meerwein rearrangement [16]. It should be noted that bromides **1a–b** were formed as racemates. Next, we performed interactions between the key intermediates **1a–b** with the corresponding saturated N-containing heterocycle. As a result of a two-step synthesis, we obtained a library of ethers **2–8 a–b** containing a bicyclic fragment and a saturated N-heterocycle, separated by linkers of different lengths (Scheme 1). As a heterocyclic component, we used pyrrolidine, piperidine, 4-methylpiperidine, 1-methylpiperazine, 1-benzylpiperazine, morpholine, and azepane. The newly synthesized compounds were characterized by ^1^H, ^13^C NMR spectroscopy, and HR-MS mass spectrometry. The spectral data confirmed all new compounds to have the expected structures and high purity.

### 2.2. Antiviral Activity Study

#### 2.2.1. Influenza Virus

Influenza viruses are divided into three types: influenza A, B, C viruses, and in particular, influenza A viruses have a long history of emergence and re-emergence. Therefore, the compounds obtained have been studied on antiviral activity against the influenza virus A/Puerto Rico/8/34 (H1N1) in madin-darby canine kidney (MDCK) cell culture (Table 1). Ribavirin and Rimantadine were used as reference compounds. According to Table 1, compounds **2a**, **3a**, **4a** and **7b** with pyrrolidine, piperidine, 4-methylpiperidine and morpholine cycle showed good antiviral activity with IC_50_ value ranging from 24.2 to 64.8 µM and low cytotoxicity. Compound **6a** with benzylpiperazine substitute demonstrated a low IC_50_ value of 3.4 µM and, at the same time, turned out to be quite toxic (CC_50_ = 33.7 µM) in MDCK cell culture.

#### 2.2.2. Filovirus Infections

Ebola virus (EBOV) and Marburg virus (MARV), members of the family Filoviridae, lead to especially dangerous viral infections because of their severe pathogenicity, potential transmission from person to person, and lack of approved vaccines or antiviral treatments. Therefore, studies using the authentic EBOV and MARV can only be performed in biosafety level 4 (BSL-4) containment laboratories. Using pseudotyped viruses allows overcoming this issue and searching for filoviruses entry inhibitors under BSL-2 containment. In this study, we created pseudotyped viruses that had the structural and enzymatic core of vesicular stomatitis virus (VSV), bearing the glycoprotein (GP) of EBOV or MARV and encoding a quantifiable reporter gene. Table 2 demonstrates the potency of all synthesized camphene derivatives **2–8 a–b** against Ebola pseudotype viruses (rVSV-ΔG-EboV-GP), Marburg pseudotype viruses (rVSV-ΔG-MarV-GP), and cytotoxicity in HEK293T cell culture. The known antidepressant sertraline was used as a reference agent since there are data confirming this drug’s ability to bind to the GP EBOV [17]. 

The structure-activity relationship (SAR) results showed that the most efficient inhibitors of rVSV-ΔG-EboV-GP were derivatives **2a** (IC_50_ = 0.12 µM), **3a** (IC_50_ = 6.3 µM), **4a** (IC_50_ = 1.3 µM), and **7a** (IC_50_ = 0.6 µM) bearing moieties of pyrrolidine, piperidine, 4-methylpiperidine, and morpholine, respectively. Wherein their analogs **2b**, **3b**, **4b**, and **7b** with linker lengths of 2, turned out to be not active against Ebola pseudotype viruses. The study of antiviral activity against the Marburg pseudotype virus showed the synthesized derivatives 2-8 a-b to have a weak inhibiting activity, with an IC_50_ (MarV-GP) value > 50 μM. The antiviral activities against viral particles pseudotyped G VSV (rVSV-ΔG-G) were determined to identify the specific inhibitory activity of the camphene derivatives **2a**, **3a**, **4a**, and **7a** toward rVSV-ΔG-EboV-GP. All compounds tested did not show the inhibiting activity, with an IC_50_ (VSV-G) value being more than 100 times higher than IC_50_ (EboV-GP).

The pseudo-infection screening identified compounds **2a**, **3a**, **4a**, and **7a** as prospective inhibitors of the entry step of the Ebola virus. Therefore these agents were assessed for their antiviral activity against the authentic filoviruses EBOV (strain Zaire) (Table 3). The anti-EBOV activity study was conducted at the State Research Center of Virology and Biotechnology Vector (SRC VB Vector) in a maximum containment facility (BSL-4). EBOV was obtained from the State Collection of Viral Infections and Rickettsioses Agents of the SRC VB Vector.

As shown in Table 3, the derivative **4a** did not show antiviral activity against EBOV and demonstrated high toxicity (50% cytotoxic concentration (CC_50_) = 40.2 µM). The derivative **2a** containing a pyrrolidine group displayed good inhibitory activity (IC_50_ = 18.3 μM) together with low toxicity (CC_50_ = 230.7 μM). The derivatives **3a** and **7a** also displayed moderate antiviral activity against EBOV but, at the same time, were quite toxic (CC_50_ = 55.9 and 57.5 μM).

#### 2.2.3. Hantavirus

Hantaviruses, members of the family Hantaviridae, are a kind of enveloped single negative chain RNA viruses spread mainly by rodents. They cause hantavirus pulmonary syndrome (HPS) or hemorrhagic fever with renal syndrome (HFRS) in humans worldwide. In total, over 40 different hantavirus species are currently known, with 22 species among them considered pathogenic for humans [19]. Currently, there is no effective treatment for either HFRS or HPS. Moreover, searching for effective inhibitors against hantaviruses diseases at the cellular level is impeded by the fact that Hantavirus tends to grow slowly even in the most susceptible cells, usually resulting in little or no cytopathic effect, nor is there a satisfactory model to track the effect on animals [20]. In the present work, we study the activity of synthesized derivatives **3a**, **2a**, and **7a** against HFRS pathogen, Hantaan virus strain 76–118, using a pseudovirus system with Hantaan virus Gn-Gc glycoprotein on its surface (rVSV-ΔG-Gn-Gc).

The results indicated that all studied compounds demonstrated high antiviral activity against the Hantaan pseudovirus with IC_50_ values between 5.0 and 14.8 µM. Taking into account the CC_50_ values, the most effective agent turned out to be derivative **7a** (selectivity index: (SI) = 78) with a morpholine fragment.

#### 2.2.4. Vaccinia Virus

Vaccinia virus (VV), a member of the *Orthopoxvirus* genus, is a live attenuated virus used worldwide by the WHO in the smallpox vaccine and widely used to find inhibitors of orthopoxviruses [21]. The antiviral activity and cytotoxicity of the synthesized derivatives against VV (Copenhagen strain) were evaluated by an adapted method in Vero cell culture [22]. As positive controls, two compounds were used: the commercially available drug Cidofovir and ST-246. The data analysis has shown that the synthesized compound had no antiviral activity against VV (Table 4).

### 2.3. Search for a Possible Target by Molecular Modelling Study

The antiviral screening showed the derivative **2a** to display broad-spectrum antiviral activity against influenza virus A/PR/8/34 (H1N1) (IC_50_ = 45.3 µM; SI = 26), EBOV (IC_50_ = 18.3 µM; SI = 12) and pseudoviruses with Hantaan virus Gn glycoprotein on its surface (IC_50_ = 9.1 µM; SI = 39). The derivative **7a** was also found to demonstrate antiviral activity. Given that the derivatives **2a** and **7a** selectively inhibited rVSV-ΔG-EboV-GP (Table 2) and rVSV-ΔG-Gn-Gc (Table 5) but did not show activity against VSV-G (IC_50_ = 123.3 and 934.9 µM, respectively), we could assume that the surface protein is likely to be the target of compounds **2a** and **7a**. The major glycoprotein on the influenza virus envelope is hemagglutinin (HA), responsible for the fusion process between viral and cellular membranes. The key protein that mediates EBOV entry into host cells is a glycoprotein (GP). A molecular modeling study was performed to characterize the details of the direct interaction of compounds **2a** and **7a** with the surface proteins of influenza virus (HA) and EBOV (GP). GP and HA are related to class I fusion proteins and have similar pre- and postfusion forms [23], while the surface glycoprotein of the Hantaan virus (Gn) is related to class II fusion proteins. We excluded Gn from the molecular modeling study because the structure and pre- and post-fusion forms of the class II fusion proteins differ from those of class I fusion proteins. 

#### 2.3.1. Binding Site Analysis

As mentioned above, surface proteins of the influenza virus and EBOV have a common mechanism of fusion of viral and cell membranes. The presence of similar heptad repeats in the stem part of HA and EBOV suggests a similar mechanism of inhibition of membrane fusion by small molecules and allows identifying similar binding sites. The HA and GP binding sites of small molecules described in the literature are characterized primarily by hydrophobic ligand-protein interactions. Thus, it seems possible to determine a “universal site” with hydrophobic characteristics present in GP and HA.

In this study, we considered the binding site of the GP EBOV located in a hydrophobic cavity between the attachment (GP1) and fusion (GP2) subunits were selected for consideration in this work. A large number of FDA-approved drugs such as toremifene [24], bepridil, paroxetine, sertraline, and benztropine [17] were shown to bind in this site (hereinafter referred to as the Toremifene-site). These drugs belong to various pharmacological groups and have different chemical structures but bind in the same cavity on the EBOV GP. Moreover, previously we have previously shown that (−)-borneol derivatives **Ia–d** (Figure 1) also bind to the Toremifene-site [12]. 

To search for a similar binding site in the HA, we examined the stem part of the HA2 subunit since there is a hydrophobic cavity favorable for the binding of small-molecule fusion inhibitors. A number of binding sites in the hydrophobic cavity have been described [25,26]. Tert-butyl hydroquinone binding site (TBHQ-site) is located at the interface between two monomers of the HA trimer [27]. Toremifene-site is characterized by a large number of hydrophobic amino acids, such as valine, leucine, and isoleucine, and it is this feature that we used as a criterion for finding a similar site in HA. We used the Binding Site Alignment procedure implemented in software Schrodinger Suite Release-2020-4 to find similar binding sites for potential surface protein inhibitors of influenza and Ebola viruses. The amino acid sequence was determined within a radius of 5 angstroms from the ligands **2a** and **7a** at the Toremifene-site of EBOV GP. Further, the program algorithm performed a pairwise superposition of several structures, followed by the determination of similar binding sites by the parameters of hydrophobicity and hydrophilicity of amino acid residues at the binding site. Thus, a similar binding site to the Toremifene-site was found in the HA2 subunit of the influenza virus. The place saturated with a large number of hydrophobic amino acids in HA is located slightly higher than the TBQH-site and near the loop connecting two α-helices of the heptad repeat. It is noteworthy that this binding site (M090) was previously discussed as a potential inhibitor [28]. Thus, we can identify two binding sites, theM090-site and Toremifene-site in HA and GP, respectively, with common structural characteristics, namely, a large number of hydrophobic interactions. For more detailed characteristics of the interaction of compounds **2a** and **7a** at potential binding sites, we present a pharmacophore model based on the assessment of the contributions of amino acid residues located in these binding sites, the pharmacophore profile of compounds **2a** and **7a**, and a pharmacophore model based on the principle of ligand and protein complementarity (see Appendix A).

#### 2.3.2. Molecular Modelling Study of Synthesized Derivatives to Binding Sites of HA and GP

The docking of the derivatives **2a** and **7a** into the Toremifene-site, TBHQ-site, and M090-site was performed, and the calculated binding energies were compared with the IC_50_ values obtained in the in vitro experiments (Figure 2). According to the in vitro experiments, the derivative **2a** demonstrated high antiviral activity against influenza virus (IC_50_ = 45.3 µM, SI = 26), and the derivative **7a** did not show a significant inhibiting effect (IC_50_ = 252 µM). The calculated binding energies of synthesized ligands **2a** and **7a** in the binding sites of HA showed that both ligands could bind into discussed sites with similar values of ΔG_bind_. The in silico docking results of ligands **2a** and **7a** in the Toremifene-site EBOV GP were in correlation with experimental in vitro data. The pyrrolidine derivative **2a** showed antiviral activity with the IC_50_ value of 18.3 µM and ΔG_bind_= −44.5 kkal/mol. Compound **7a** exhibited higher affinity (ΔG_bind_= −51.4 kkal/mol) and virus-inhibiting activity was better (IC_50_ = 5.6 µM) than that of compound **2a**.

Overall, the binding of ligands **2a** and **7a** and surface proteins of the influenza and Ebola viruses were characterized mainly by hydrophobic contacts, where 1,7,7-trimethylbicyclo(2.2.1)heptan scaffold formed hydrophobic interactions with valine, leucine, and isoleucine. In the Toremifene-site, compounds **2a** and **7a** bind better than in TBHQ-site and M090-site, probably due to the toremifene site being more hydrophobic.

## 3. Conclusions

A series of novel camphene derivatives **2–8 a–b** have been synthesized, characterized, and evaluated as antiviral compounds. All compounds underwent a broad evaluation of antiviral activity against a panel of DNA and RNA viruses, i.e., vaccinia virus, influenza virus A/PR/8/34 (H1N1), Ebola pseudotype viruses, Marburg pseudotype viruses, authentic EBOV (strain Zaire), and the Hantaan pseudotype virus (strain 76–118). According to the antiviral study results, the derivatives **2–8 a–b** did not show activity against the vaccinia virus, while several derivatives did demonstrate significant antiviral potency against other viruses. According to the antiviral study results, the derivatives **2–8 a–b** did not show activity against the vaccinia virus, while several derivatives did demonstrate significant antiviral potency against other viruses. Hence, surface proteins can be suggested to be a likely molecular target of these derivatives. In the molecular modeling study, we compared the binding sites of the potential entry of inhibitors, assuming the mechanisms of fusion of type I surface proteins to be similar, and estimated the affinity of ligands **2a** and **7a** for them by calculating the binding energy of the ligand and the protein in the ligand-protein complex. According to the in silico results, a bicyclic scaffold provides efficient binding to the hydrophobic part of the binding site of the surface proteins under consideration, and protonated nitrogen provides electrostatic interactions. Further search for new analogs, including these two structural fragments, may lead to discovering a new inhibitor targeting the membrane fusion stage and possessing a broad spectrum of antiviral activity.

## 4. Materials and Methods

### 4.1. Chemistry

#### 4.1.1. General Information

Reagents and solvents were purchased from commercial suppliers and used as received. Dry solvents were obtained according to standard procedures. Reactions monitoring, the content of the compounds in fractions during chromatography, and the purity of the target compounds were determined using 7890A gas chromatograph (Agilent Tech., Santa Clara, CA, USA) with an Agilent 5975C quadrupole mass spectrometer as the detector; HP-5 capillary column, He as carrier gas (flow rate 2 mL/min, flow division 99:1). ^1^H and ^13^C NMR spectra were recorded on Bruker spectrometers, including an AV-300 instrument at 300.13 MHz (^1^H) and 75.47 MHz (^13^C), and an AV-400 instrument at 400.13 MHz (^1^H) and 100.61 MHz (^13^C), and a DRX-500 instrument at 500.13 MHz (^1^H) and 125.76 MHz (^13^C) in CDCl_3_; chemical shifts δ were reported in ppm relative to residual CHCl_3_ (d(CHCl_3_) 7.24, d(CDCl_3_) 76.90 ppm), J in Hz. High-resolution mass spectra (HRMS) were obtained with a DFS Thermo Scientific mass spectrometer in a full scan mode (0–500 *m*/*z*, 70 eV electron impact ionization, direct sample administration).

#### 4.1.2. Synthesis of 2-(2-bromoethoxy)-1,7,7-trimethylbicyclo(2.2.1)eptanes **1a** and 2-(3-bromopropoxy)-1,7,7-trimethylbicyclo(2.2.1)heptanes **1b**

A mixture of 2-bromoethanol (1.1 eqv.) or 3-bromopropan-1-ol (1.1 eqv.), and the excess clay K-10 and (±)-camphene (1 eqv.) in CH_2_Cl_2_ were stirred at room temperature for 24 h. The precipitate of clay was filtered off, and the solution was washed with brine, extracted with CH_2_Cl_2_, and dried over anhydrous Na_2_SO_4_. After complete solvent evaporation, the residue was subjected to silica gel column chromatography eluting with hexane to give **1a** or **1b**.

(±)-*2-(2-Bromoethoxy)-1,7,7-trimethylbicyclo(2.2.1)heptanes* (**1a**), *Colorless oil*, Yield 75%; ^1^H-NMR (CDCl_3_) δ: 0.78 (3H, s), 0.87 (3H, s), 0.96 (3H, s), 0.94–0.98 (2H, m), 1.43–1.78 (5H, m), 3.19–3.23 (1H, m), 3.37–3.42 (1H, m), and 3.59–3.71 (2H, m). ^13^C-NMR (CDCl3) δ (ppm): 11.7 q, 20.0 q, 20.1 q, 27.1 t, 31.1 t, 34.2 t, 38.5 t, 44.9 d, 46.3 s, 49.2 s, 69.2 t, and 87.5 d. HRMS: calc. for C_12_H_21_BrO [M]^+^ 260.0770. Found 260.0761.

(±)-*2-(3-Bromopropoxy)-1,7,7-trimethylbicyclo(2.2.1)heptanes* (**1b**), *Colorless oil*, Yield 62%; ^1^H-NMR (CDCl_3_) δ (ppm): 0.78 (3H, s), 0.85 (3H, s), 0.93 (3H, s), 0.94–0.98 (2H, m), 1.42–1.75 (5H, m), 1.94–2.08 (2H, m), 3.13–3.17 (1H, m), 3.28–3.34 (1H, m), and 3.45–3.54 (3H, m). ^13^C-NMR (CDCl_3_) δ (ppm): 11.7 q, 20.0 q, 20.1 q, 27.2 t, 31.3 t, 33.1 t, 34.3 t, 38.3 t, 44.9 d, 46.3 s, 49.1 s, 65.9 t, and 87.1 d. HRMS: calc. for C_13_H_23_BrO [M]^+^ 274.0927. Found 274.0923. 

#### 4.1.3. General Procedure for Synthesis of Derivatives **2–8 a–b**

A mixture of **1a** (1 eqv.) or **1b** (1 eqv.), K_2_CO_3_ (2 eqv.) and corresponding secondary amine (1.2 eqv.) in CH_3_CN was stirring at room temperature for 12 h. After that time, the reaction mixture was evaporated, washed with brine, and extracted with diethyl ether (3 × 10 mL). The combined organic extracts were dried over anhydrous Na_2_SO_4_. After complete evaporation of the solvent, the residue was subjected to column chromatography eluting with hexane-ethyl acetate.

*(±)-1-(2-(1,7,7-Trimethylbicyclo(2.2.1)heptan-2-yloxy)ethyl)pyrrolidine* (**2a**). The compound was obtained from the reaction of **1a** with pyrrolidine as pale yellow oil in yield 63%. ^1^H-NMR (CDCl_3_) δ (ppm): 0.76 (3H, s), 0.83 (3H, s), 0.92 (3H, s), 0.93–0.96 (2H, m), 1.41–1.56 (2H, m), 1.58–1.74 (5H, m), 2.35–2.42 (6H, m), 3.08–3.16 (1H, m), 3.20–3.28 (1H, m), and 3.33–3.46 (1H, m). ^13^C-NMR (CDCl_3_) δ (ppm): 11.7 q, 20.0 q, 20.1 q, 27.0 t, 27.2 t, 34.3 t, 38.4 t, 44.9 d, 46.2 s, 49.0 s, 53.6 t, 56.1 t, 66.9 t, and 86.8 d. HRMS: calc. for C_17_H_31_NO_2_ [M]^+^ 281.2349; found 281.2357.

*(±)-4-(3-(1,7,7-trimethylbicyclo(2.2.1)heptan-2-yloxy)propyl)morpholine* (**2b**). The compound was obtained from the reaction of **1b** with pyrrolidine as pale yellow oil in yield 49%. ^1^H-NMR (CDCl_3_) δ (ppm): 0.76 (3H, s), 0.83 (3H, s), 0.93 (3H, s), 0.91–0.96 (2H, m), 1.39–1.78 (10H, m), 2.39–2.55 (6H, m), 3.09–3.16 (1H, m), 3.22–3.31 (1H, m), and 3.37–3.45 (1H, m). ^13^C-NMR (CDCl_3_) δ (ppm): 11.7 q, 20.0 q, 20.1 q, 23.2 t, 27.1 t, 29.3 t, 34.2 t, 38.4 t, 44.8 d, 46.3 s, 48.9 s, 53.6 t, 54.0 t, 67.1 t, and 86.6 d. HRMS: calc. for C_16_H_28_NO [M-CH_3_] ^+^ 250.2165; found 250.2163.

*(±)-1-(2-(1,7,7-Trimethylbicyclo(2.2.1)heptan-2-yloxy)ethyl)piperidine* (**3a**). The compound was obtained from the reaction of **1a** with piperidine as pale yellow oil in yield 53%. ^1^H-NMR (CDCl_3_) δ (ppm): 0.74 (3H, s), 0.82 (3H, s), 0.91 (3H, s), 0.91–0.94 (2H, m), 1.33–1.73 (11H, m), 2.34–2.45 (4H, br s), 2.45–2.50 (2H, m), 3.10–3.15 (1H, m), 3.32–3.39 (1H, m), and 3.49–3.55 (1H, m). ^13^C-NMR (CDCl_3_) δ (ppm): 11.7 q, 19.9 q, 20.0 q, 24.0 t, 25.8 t, 27.1 t, 34.3 t, 38.3 t, 44.9 d, 46.2 s, 48.9 s, 54.8 t, 58.6 t, 67.3 t, 87.2 d.

*(±)-1-(3-(1,7,7-Trimethylbicyclo(2.2.1)heptan-2-yloxy)propyl)piperidine* (**3b**). The compound was obtained from the reaction of **1b** with piperidine as pale yellow oil in yield 44%. ^1^H-NMR (CDCl_3_) δ (ppm): 0.77 (3H, s), 0.84 (3H, s), 0.92 (3H, s), 0.91–0.97 (2H, m), 1.35–1.73 (13H, m), 2.28–2.39 (6H, m), 3.10–3.15 (1H, m), 3.19–3.27 (1H, m), and 3.35–3.43 (1H, m). ^13^C-NMR (CDCl_3_) δ (ppm): 11.7 q, 20.1 q, 20.2 q, 24.3 t, 25.8 t, 27.2 t, 27.3 t, 34.4 t, 38.5 t, 45.0 d, 46.3 s, 49.1 s, 54.4 t, 56.5 t, 67.3 t, and 86.8 d. HRMS: calc. for C_17_H_30_NO [M-2H]^+^ 264.2322; found 264.2319.

*(±)-4-Methyl-1-(2-(1,7,7-trimethylbicyclo(2.2.1)]heptan-2-yloxy)ethyl)piperidine* (**4a**). The compound was obtained from the reaction of **1a** with 4-methylpiperidine as pale yellow oil in yield 51%. ^1^H-NMR (CDCl_3_) δ (ppm): 0.74 (3H, s), 0.81 (3H, s), 0.85 (3H, d, J = 6.5 Hz), 0.90 (3H, s), 0.90–0.95 (2H, m), 1.07–1.35 (3H, m), 1.37–1.73 (7H, m), 1.89–2.03 (2H, m), 2.41–2.53 (2H, m), 2.45–2.50 (2H, m), 2.78–2.95 (2H, m), 3.09–3.15 (1H, m), 3.28–3.39 (1H, m), and 3.45–3.55 (1H, m). ^13^C-NMR (CDCl_3_) δ (ppm): 11.8 q, 20.0 q, 20.1 q, 21.8 q, 27.2 t, 30.4 d, 34.3 t, 38.3 t, 44.9 d, 46.2 s, 49.0 s, 54.3 t, 54.4 t, 58.3 t, 67.5 t, and 87.3 d. HRMS: calc. for C_18_H_32_NO [M-H]^+^ 278.2478; found 278.2480.

*(±)-4-Methyl-1-(3-(1,7,7-trimethylbicyclo(2.2.1)heptan-2-yloxy)propyl)piperidine* (**4b**). The compound was obtained from the reaction of **1b** with 4-methylpiperidine as pale yellow oil in yield 68%. ^1^H-NMR (CDCl_3_) δ (ppm): 0.75 (3H, s), 0.83 (3H, s), 0.88 (3H, d, J = 6.5 Hz), 0.92 (3H, s), 0.92–0.96 (2H, m), 1.14–1.34 (3H, m), 1.38–1.72 (9H, m), 1.81–1.89 (2H, m), 2.31–2.36 (2H, m), 2.81–2.89 (2H, m), 3.09–3.15 (1H, m), 3.20–3.27 (1H, m), and 3.35–3.43 (1H, m). ^13^C-NMR (CDCl_3_) δ (ppm): 11.7 q, 20.0 q, 20.1 q, 21.7 q, 27.2 t, 30.5 d, 33.8 t, 34.3 t, 38.4 t, 44.9 d, 46.2 s, 49.0 s, 53.8 t, 56.1 t, 67.1 t, and 86.8 d. HRMS: calc. for C_19_H_33_NO [M-2H]^+^291.2557; found 291.2558.

*(±)-1-Methyl-4-(2-(1,7,7-trimethylbicyclo(2.2.1)heptan-2-yloxy)ethyl)piperazine* (**5a**). The compound was obtained from the reaction of **1a** with 1-methylpiperazine as pale yellow oil in yield 58%. ^1^H-NMR (CDCl_3_) δ (ppm): 0.76 (3H, s), 0.83 (3H, s), 0.91 (3H, s), 0.91–0.97 (2H, m), 1.37–1.74 (5H, m), 2.25 (3H, s), 2.32–2.60 (10H, m), 3.10–3.17 (1H, m), 3.31–3.42 (1H, m), and 3.48–3.59 (1H, m). ^13^C-NMR (CDCl_3_) δ (ppm): 11.8 q, 20.0 q, 20.1 q, 27.2 t, 34.3 t, 38.3 t, 44.9 d, 45.9 q, 46.3 s, 49.0 s, 53.6 t, 55.1 t, 58.0 t, 67.4 t, and 86.4 d. HRMS: calc. for C_17_H_32_N_2_O [M]^+^ 280.2509; found 280.2505.

*(±)-1-Methyl-4-(3-(1,7,7-trimethylbicyclo(2.2.1)heptan-2-yloxy)propyl)piperazine* (**5b**). The compound was obtained from the reaction of **1b** with 1-methylpiperazine as pale yellow oil in yield 53%. ^1^H-NMR (CDCl_3_) δ (ppm): 0.74 (3H, s), 0.82 (3H, s), 0.91 (3H, s), 0.91–0.95 (2H, m), 1.36–1.70 (7H, m), 2.24 (3H, s), 2.27–2.60 (10H, m), 3.07–3.13 (1H, m), 3.18–3.27 (1H, m), and 3.36–3.44 (1H, m). ^13^C-NMR (CDCl_3_) δ (ppm): 11.7 q, 20.0 q, 20.1 q, 27.2 t, 27.3 t, 34.3 t, 38.4 t, 44.9 d, 45.9 q, 46.2 s, 49.0 s, 53.1 t, 55.0 t, 55.7 t, 67.0 t, and 86.7 d. HRMS: calc. for C_18_H_34_N_2_O [M]^+^ 294.2666; found 294.2664.

*(±)-1-Benzyl-4-(2-(1,7,7-trimethylbicyclo(2.2.1)heptan-2-yloxy)ethyl)piperazine* (**6a**). The compound was obtained from the reaction of **1a** with 1-benzylpiperazine as pale yellow oil in yield 43%. ^1^H-NMR (CDCl_3_) δ (ppm): 0.76 (3H, s), 0.83 (3H, s), 0.91 (3H, s), 0.92–0.96 (2H, m), 1.41–1.72 (7H, m), 2.17–2.63 (10H, m), 3.11–3.16 (1H, m), 3.35–3.41 (1H, m), 3.48 (2H, s), 3.51–3.59 (1H, m), and 7.27–7.32 (5H, m). ^13^C-NMR (CDCl_3_) δ (ppm): 11.8 q, 20.1 q, 20.1 q, 27.2 t, 34.3 t, 38.3 t, 44.9 d, 46.3 s, 49.0 s, 52.9 t, 53.6 t, 58.1 t, 67.4 t, 86.4 d, 126.9 d, 128.1 d, 129.1 d, and 138.0 s. HRMS: calc. for C_23_H_36_N_2_O [M]^+^356.2822; found 356.2819.

*(±)-1-Benzyl-4-(3-(1,7,7-trimethylbicyclo(2.2.1)heptan-2-yloxy)propyl)piperazine* (**6b**). The compound was obtained from the reaction of **1b** with 1-benzylpiperazine as pale yellow oil in yield 37%. ^1^H-NMR (CDCl_3_) δ (ppm): 0.77 (3H, s), 0.83 (3H, s), 0.92 (3H, s), 0.92–0.96 (2H, m), 1.40–1.55 (2H, m), 1.61–1.71 (5H, m), 2.32–2.55 (10H, m), 3.09–3.14 (1H, m), 3.21–3.27 (1H, m), 3.36–3.42 (1H, m), 3.48 (2H, s), and 7.27–7.30 (5H, m). ^13^C-NMR (CDCl_3_) δ (ppm): 11.7 q, 20.0 q, 20.1 q, 27.2 t, 27.3 t, 34.3 t, 38.4 t, 44.9 d, 45.9 q, 46.2 s, 49.0 s, 52.9 t, 53.0 t, 55.7 t, 63.0 t, 67.0 t, 86.7 d, 126.9 d, 128.0 d, 129.1 d, and 137.9 s. HRMS: calc. for C_24_H_38_N_2_O [M]^+^370.2979; found 370.2985.

*(±)-4-(2-(1,7,7-Trimethylbicyclo(2.2.1)heptan-2-yloxy)ethyl)morpholine* (**7a**). The compound was obtained from the reaction of **1a** with morpholine as pale yellow oil in yield 59%. ^1^H-NMR (CDCl_3_) δ (ppm): 0.74 (3H, s), 0.81 (3H, s), 0.89 (3H, s), 0.91–0.94 (2H, m), 1.38–1.53 (2H, m), 1.56–1.63 (2H, m), 1.65–1.71 (1H, m), 2.41–2.51 (6H, m), 3.08–3.14 (1H, m), 3.32–3.37 (1H, m), 3.48–3.54 (1H, m), and 3.59–3.67 (4H, m). ^13^C-NMR (CDCl_3_) δ (ppm): 11.7 q, 19.9 q, 20.0 q, 27.1 t, 34.2 t, 38.2 t, 44.8 d, 46.1 s, 48.9 s, 54.0 t, 58.4 t, 66.8t, 67.3 t, and 87.3 d.

*(±)-4-(3-(1,7,7-trimethylbicyclo(2.2.1)heptan-2-yloxy)propyl)morpholine* (**7b**). The compound was obtained from the reaction of **1b** with morpholine as pale yellow oil in yield 79%. ^1^H-NMR (CDCl_3_) δ (ppm): 0.76 (3H, s), 0.83 (3H, s), 0.92 (3H, s), 0.93–0.96 (2H, m), 1.41–1.56 (2H, m), 1.58–1.74 (5H, m), 2.35–2.42 (6H, m), 3.08–3.16 (1H, m), 3.20–3.28 (1H, m), 3.33–3.46 (1H, m), and 3.60–3.74 (4H, m). ^13^C-NMR (CDCl_3_) δ (ppm): 11.7 q, 20.0 q, 20.1 q, 27.0 t, 27.2 t, 34.3 t, 38.4 t, 44.9 d, 46.2 s, 49.0 s, 53.6 t, 56.1 t, 66.9 t, and 86.8 d. HRMS: calc. for C_17_H_31_NO_2_ [M]^+^ 281.2349; found 281.2357.

*(±)-1-(2-(1,7,7-Trimethylbicyclo(2.2.1)heptan-2-yloxy)ethyl)azepane* (**8a**). The compound was obtained from the reaction of **1a** with azepane as colorless oil in yield 60%. ^1^H-NMR (CDCl_3_) δ (ppm): 0.77 (3H, s), 0.85 (3H, s), 0.93 (3H, s), 0.92–0.96 (2H, m), 1.38–1.73 (13H, m), 2.61–2.70 (6H, m), 3.10–3.17 (1H, m), 3.31–3.38 (1H, m), and 3.46–3.55 (1H, m). ^13^C-NMR (CDCl_3_) δ (ppm): 11.8 q, 20.0 q, 20.1 q, 26.8 t, 27.2 t, 27.7 t, 34.3 t, 38.3 t, 44.9 d, 46.3 s, 49.0 s, 55.8 t, 57.4 t, 67.7 t, and 87.3 d. HRMS: calc. for C_18_H_31_NO [M-2H]^+^277.2400; found 277.2404.

*(±)-1-(3-((1,7,7-Trimethylbicyclo(2.2.1)heptan-2-yloxy)propyl)azepane* (**8b**). The compound was obtained from the reaction of **1b** with azepane as colorless oil in yield 47%. ^1^H-NMR (CDCl_3_) δ (ppm): 0.76 (3H, s), 0.84 (3H, s), 0.92 (3H, s), 0.92–0.96 (2H, m), 1.39–1.73 (15H, m), 2.47–2.52 (2H, m), 2.56–2.60 (4H, m), 3.10–3.14 (1H, m), 3.19–3.27 (1H, m), and 3.36–3.43 (1H, m). ^13^C-NMR (CDCl_3_) δ (ppm): 11.7 q, 20.0 q, 20.1 q, 26.8 t, 27.2 t, 27.5 t, 27.6 t, 34.3 t, 38.4 t, 44.9 d, 46.2 s, 49.0 s, 55.2 t, 58.2 t, 67.0 t, and 86.7 d. HRMS: calc. for C_19_H_33_NO [M-2H]^+^ 291.2557; found 291.2559.

### 4.2. Biological Studies

#### 4.2.1. Evaluation of the Anti-Vaccinia Virus Activity

The vaccinia virus (strain Copenhagen) was obtained from the State collection of pathogens of viral infections and rickettsioses of the SRC VB Vector (Koltsovo, Novosibirsk region, Russia). The virus was grown in a Vero cell culture in DMEM medium (BioloT, St.-Petersburg, Russia). The concentration of viruses in the culture fluid was determined by the method of plaques by titration of the samples in Vero cell culture, calculated and expressed in decimal logarithms of plaque-forming units per ml (log_10_ PFU per mL). The concentration of the virus in the samples used in this work was from 5.6 to 6.1 log_10_ PFU per ml. For the determination of the cytotoxicity and antiviral activities of agents against the Vaccinia virus, we used the adapted method previously described [22].

#### 4.2.2. Evaluation of the Anti-Influenza Virus Activity

Influenza virus A/Puerto Rico/8/34 (H1N1) was obtained from the collection of viruses of St Petersburg Pasteur Institute, Russia, and used in the study. Prior to the experiment, the virus was propagated in the allantoic cavity of 10–12 days-old chicken embryos for 48 h at 36 °C. Infectious titer of the virus was determined in MDCK (madin-darby canine kidney) cells (ATCC # CCL-34) in 96-wells plates in alpha-MEM medium (Biolot, St.-Petersburg, Russia). The cytotoxicity assay and virus inhibition assay were performed as previously described [29].

#### 4.2.3. Evaluation of the Anti-Filovirus Activity

VSV pseudotyped viruses rVSV-ΔG-EboV-GP and rVSV-ΔG-MarV-GP were created as described previously [12]. HEK293FT cells were grown in a T75 culture vial were transfected with a pGPE (or pGPM) plasmid using the CaCl_2_ method (23 mg of plasmid per cell monolayer in a T75 culture vial). After 16 h, the culture medium was replaced, and a suspension of VSV of the firefly luciferase gene pseudotyped by surface VSV glycoprotein G (rVSV-ΔG-G) was added to the cells [30]. After 6 h, the cells were washed, and the medium was exchanged with a fresh medium. Pseudoviruses were harvested after 48 h by filtering the culture medium through a 0.45-mm filter after centrifugation to remove cell debris. Pseudoviruses were stored at −80 °C, and their functional activity was determined using a HEK293T cell culture, with the luminescence level recorded using a Stat Fax 4400 luminometer.

Viral entry inhibition assays were performed using HEK293T cells. Cells were seeded in 96-well plates (100 mL at a density of 10^5^ cells per ml) one day before analysis. The following day, potential inhibitor compounds were titrated in 96-well round-bottom plates at a dilution of 1:4. Next, 10-mL aliquots of pseudoviruses (10^5^ RLU) were added to each well, and the mixtures were incubated for 1 h in a CO_2_ incubator at 37 °C under an atmosphere with 5% CO_2_. After incubation, an aliquot of each mixture was added to a monolayer of cells. As a negative control, cells were treated with the same volume of the medium as that used for the pseudovirus compound mixture was added to cells. After incubating in a CO_2_ incubator for 48 h, the luminescence level was measured using a Stat Fax 4400 plate luminometer. To this end, the medium was carefully removed from the wells, the cells were washed with 100 mL/well of PBS, and then 25 mL of Luciferase Cell Culture Lysis Buffer (Promega) was added. After 5 min, 50 mL of Luciferase Assay Reagent (Promega) were added. The percentage of inhibition was evaluated by determining the degree of decrease in luminescence in the wells with the compounds relative to the control wells (cells with the virus without compounds). All test compounds were dissolved in dimethylsulfoxide (DMSO) at a concentration of 10 mg/mL and used at a final concentration of 375 to 0.02 mg/mL.

MTT reduction was used to study the cytotoxicity of the compounds [31]. Briefly, a series of two-fold dilutions of each compound (15.6–1000 µM) in 10% DMEM were prepared in 96-well plates. HEK293T cells (100 mL at a density of 10^5^ cells per ml) were added and incubated for 48 h at 37 C in 5% CO_2_. Then 20 mL (1/10 vol) of a solution of 3-(4,5-dimethylthiazolyl-2)-2,5-diphenyltetrazolium bromide (Sigma) (5 mg/mL) in phosphate-buffered saline was added to each well. After 2 h of incubation, the solution was removed from wells, and DMSO (100 mL per well) was added to dissolve the formazan crystals. The optical density of cells was then measured on a Model 680 Microplate (Bio-Rad) Reader at 535 nm and plotted against the concentration of the compounds. Each concentration was tested in three parallels. The 50% cytotoxic dose (CC_50_) of each compound was calculated from the data obtained.

Anti-EBOV assays were conducted in BSL-4 conditions at SRC VB Vector. EBOV (strain Zaire) was obtained from the State Collection of Viral Infections, and Rickettsioses Agents of SRC VB Vector were suspended in the culture supernatant. Titer EBOV was 4.5 ± 0.75 lgTCD_50_/_mL_ (decimal logarithm of 50% tissue cytopathic dose). Vero cell cultures were seeded into 96-well plates and were grown into confluent monolayers. Series of 3-fold dilutions of each compound (previously dissolved in 100% DMSO) in DMEM medium were prepared, starting with 300 mg/mL. Six decreasing concentrations were prepared from each inhibitor. The 100 mL/well of compounds was added. EBOV was used at a multiplicity of infections of 0.01 (equivalent to a dose of 100 TCID_50_ per well). For the treatment assay, 100 µL of diluted virus (100 TCID_50_) were added. For cytotoxicity assay, 100 mL of DMEM medium were added to each well. After incubation at 37 °C and 5% CO_2_ for 10 days, neutral red was then added. The absorbance was measured at 490 nm using a microplate reader (Thermo Scientific Multiskan FC). The 50% cytotoxicity concentration (CC_50_) and 50% inhibitory concentration (IC_50_) were calculated using SOFTmax PRO 4.0 with the 4-parameter analysis method.

#### 4.2.4. Evaluation of the Anti-Hantavirus Activity

To generate Hantaan glycoprotein pseudotyped VSVs containing the firefly luciferase gene, HEK293T cells grown in a T75 were transfected with a pcDNA3.1-Han plasmid using Lipofectamine 3000 (ThermoFisher, Waltham, MA, USA) in accordance with the manufacturer’s protocol. After 48 h the culture medium was replaced, and a suspension of VSV of the firefly luciferase gene pseudotyped by surface VSV glycoprotein G (rVSV-ΔG-G) was added to the cells. After 6 h, the cells were washed, and the medium was exchanged with fresh medium. Pseudoviruses were harvested after 48 h by filtering the culture medium through a 0.45-mm filter after centrifugation to remove cell debris. Pseudoviruses were stored at −80 °C, and their functional activity was determined using a HEK293T cell culture, with the luminescence level recorded using a Stat Fax 4400 luminometer.

To analyze the effect of the compounds, 100 μg of compound was added to the upper well of a 96-well flat-bottom culture plate and titrated in 1:4 steps. Thus, the final concentration of compounds was from 375 μg/mL to 0.02 μg/mL in DMEM growth medium. Then to each well was added a suspension of pseudoviral particles, 25 μL (500,000 RLU), and incubated for 1 h in a CO_2_ incubator (37 °C, 5% CO_2_). At the end of the time, 100 μL of HEK293T cell suspension (100,000/mL) was added to the wells and incubated for 48 h in a CO_2_ incubator. Then, the growth medium was removed from the wells of the plate, and the wells were washed with 100 μL PBS (Phosphate-buffered saline, Rosmedbio LLC). Then added 25 μL of 1 × lysis buffer (Promega) and incubated for 10 min at room temperature. The lysed cells were suspended, and 20 μL of the suspension was transferred to an optical plate, adding 35 μL of LAR (Luciferase Assay System, Promega). The luminescence signal was recorded using a luminometer (LuMate). Untreated cells (positive) and cells treated with pseudoviruses (negative) were used as controls. The level of neutralization was determined by the decrease in the relative level of luminescence relative to the positive control. The IC_50_ value (50% cytotoxic concentration) was taken to be the concentration of the substance at which the luminescence level decreases by 50%.

### 4.3. Molecular Docking Study

Protein geometric parameters obtained by crystallographic method downloaded from a non-commercial database Protein Data Bank [32]. For the theoretical study of surface proteins, the hemagglutinin of the influenza virus strain A/H1N1-PR (PDB code 1RU7) [33] and Zaire ebolavirus glycoprotein stain Mayinga-76 (PDB code 5JQ7) [24] were considered.

The geometric parameters of the ligands (two stereoisomers of the compound **7a**, taking into account its possible protonated forms at the nitrogen atom in the morpholine fragment) were optimized by a semi-empirical method [34] using the MOPAC2016 software. In order to determine the ratio of protonated and non-protonated forms of compound **7a** by the method of quantum chemistry, the pKa value was estimated by the method of quantum chemistry. The calculations were carried out in the Jaguar program [35] using the previously described methodology [36]. The model structures of proteins were prepared accordingly for the calculations: hydrogen atoms were added and minimized, missing amino acid side chains were added, bond multiplicities were restored, solvent molecules and excess low-molecular compounds were removed, all structures were optimized in a limited force field OPLSe3 [37] at low pH values of the medium.

The docking procedures were carried out using the Schrodinger Suite: Release 2020-4, Schrödinger, LLC, New York, NY, 2020 program packages. For calculations, the monomeric form of surface proteins was considered. Derivative 2a, taking into account its stereoisomers and protonated states, was docked using the forced ligand positioning protocol (IFD) with the following conditions: flexible protein and ligand; grid matrix size of 15 Å; and amino acids (within a radius of 5 Å from the ligand) restrained and optimized, taking into account the influence of the ligand. Docking solutions were ranked by evaluating the following calculation parameters: docking score (based on GlideScore minus penalties), ligand efficiency (LE), and parameter of model energy value (Emodel), including GlideScore value, energy unrelated interactions, and the parameters of the energy spent on the formation of the laying of the compound in the binding site. For the most advantageous positions, binding energies (ΔG_MM-GBSA_) ligand-protein complexes were estimated using the variable-dielectric generalized Born model, which incorporates residue-dependent effects. Water was used as the solvent.

## Data Availability

The data presented in this study are available in this article or Appendix A.

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
