# Peer review of "Synthesis and Antiviral Activity of Camphene Derivatives against Different Types of Viruses"

_molecules, 2021, doi:10.3390/molecules26082235_

Round 1

Reviewer 1 Report

The manuscript "Synthesis and antiviral activity of camphene derivatives against different types of viruses" by A. S. Sokolova et al. describes the synthesis and the in vitro antiviral evaluation for a series of 16 camphene derivatives against influenza virus (H1N1), 2 pseudotype filoviruses (EBOV and MARV), a pseudovirus system with the Hantaan virus and Vaccinia virus (for which the compounds displayed no activity, though).

Overall the work has been carried out with high scientific standards and the presentation of the results is nice. The synthesis of compounds is described and documented appropriately with the 1H and 13C NMR spectra. The antiviral assays have been performed according to published procedures or methodologies described sufficiently and the antiviral activity is compared with their cytotoxicity in HEK293T cells. A nice supplement is presented by molecular docking calculations, which indicates putative binding sites in a very comprehensive way of presentation.

Irrespective of the antiviral activity displayed by the new compounds, which in some cases is competitive to known antiviral agents, this work presents a nice paradigm of a complete and comprehensive study comprising the rational design strategy, the syntheisis and antiviral evaluation, in addition to a structure-based target identification hypothesis. I believe this article will be very interesting for the readers of Molecules and will set a high quality standard of presentation as well.

As I have no major suggestions to improve the quality of the manuscript, therefore I suggest its publication to Molecules at its present form.

Author Response

We much appreciate the reviewer’s agreed to review this manuscript and for highly appreciating the research presented. We made some corrections to improve the manuscript.

Reviewer 2 Report

This article by Salakhutdinov et al. presents the synthesis of camphene derivatives and their biological assessment for the antiviral activity against a panel of pathogenic viruses including influenza virus, Ebola virus and the  Hantaan virus. Although the development of broad spectrum antiviral compounds is of potential interest for the readers of Molecules, I feel that this paper lacks the novelty and originality that are required for this Journal. In fact, the compounds herein presented are basically very very similar to compounds previously developed by the authors and reported in another paper (“Sokolova, A. S., Kovaleva, K. S., Yarovaya, O. I., Bormotov, N. I., Shishkina, L. N., Serova, O. A., Sergeev, A. A., Agafonov, A. P., Maksuytov, R. A., Salakhutdinov, N. F., Arch. Pharm. 2021, e2100038)”. Also, the authors claim that “surface proteins of the viruses required for the fusion process between viral and cellular membranes are the target of compound 2а,” just by performing a molecular modelling study. Overall, I feel that this statement should be supported by more in depth experiments and that the synthesis of analogues which better fit within the binding surface of these protein targets should be also prepared to confirm this observation. Overrl, I feel that  this study is too premature and should not be published in this journal in the current state.

Author Response

We are sorry to hear that the Reviewer did not find any novelty and originality in these studies. All presented compounds are new and antiviral activity was also investigated for the first time. Despite the fact that these compounds are similar to those published earlier (European Journal of Medicinal Chemistry 207 (2020) 112726), they have completely different properties. For example, we have found that previously published esters are prone to hydrolysis in a biological environment [Journal of Pharmaceutical and Biomedical Analysis, Biostability study, quantitation method and preliminary pharmacokinetics of a new antifilovirus agent based on borneol and 3-(piperidin-1-yl)propanoic acid, published in progress] while the presented compounds are stable.

Moreover, our assumption regarding the target is based not only on the data of molecular modeling. In vitro experiments using pseudoviral particles also suggest a surface protein as a potential target. As shown in Tables 2 and 4, compound 2a exhibited significant virus-inhibiting activity against Ebola pseudotype viruses (rVSV-ΔG-EboV-GP) and pseudovirus system with Hantaan virus Gn-Gc glycoprotein on its surface (rVSV-ΔG-Gn-Gс). To determine the specific inhibitory activity of the compound 2a toward rVSV-ΔG-EboV-GP and rVSV-ΔG-Gn-Gс, antiviral activities against viral particles pseudotyped G VSV (rVSVΔG-G) were determined (Table 2). IC50 value for rVSV ΔG-G for compound 2a is significantly higher than for pseudoviruses rVSV-ΔG-EboV-GP and rVSV-ΔG-Gn-Gс. Thus, based on in vitro results and molecular modeling data, we assume that a surface protein is a likely target.

Reviewer 3 Report

The paper presents synthesis, characterization and antiviral activity of novel camphene derivatives.

The molecules are new and fully characterized. The chemistry strategy is simple and the reaction yields are good. The antiviral activity of compounds is good.

The work is of general interest, is well planned and described. In my opinion the paper is worth studying and the manuscript contains enough original and interesting material.

Minor corrections which can be made by the copy editor appears to be the only problem.

Author Response

We much appreciate the reviewer’s agreed to review this manuscript. We made some corrections to improve the manuscript.

Reviewer 4 Report

A. S. Sokolova and co-workers report a two-step synthesis of 14 N-heterocyclic camphene derivatives as well as some in vitro studies against a panel of viruses (e.g. Influenza, Ebola, and Hantavirus). I consider all efforts behind viral diseases are welcome nowadays, and such results worth of publication. In this context, I found enough positive aspects making this article suitable for publication in Molecules. Introduction and background parts show suitably the context of this work, references cited are enough and pertinent, chemical yields are moderate, CC50 and IC50 values are moderate but enough to make good SAR analyses, docking studies were well performed, experimental and supplementary information seem to be Okay, conclusions are well supported. However, some minor points should be revised before publication, as follows:

- Chemical yields should be placed into Scheme 1

- Figure 2 has a lack of sharpness. It should be enhanced.

- For the 13C NMR, the peaks should have only one digit after dot

- The formula for compound 1b, the numbers should be in subscript (pg 10, ln 338), also for CDl3, lines 335-336 (pg 10). The number 13 for 13C NMR should be in superscript (pg 10, ln 336).

- I strongly recommend completing the work making a QSAR and a pharmacophore model.

- It would have been good making tests against SARS-CoV-2 strains.

Author Response

We appreciate the thoughtful comments provided by the Reviewer

  1. We have placed chemical yields in scheme 1.
  2. We sharpened the Figure 2.
  3. We have corrected the description of the 13C NMR spectra.
  4. We have made a pharmacophore model based on the assessment of the contributions of amino acid residues located in these binding sites, the pharmacophore profile of compounds 2a and 7a, and a pharmacophore model based on the principle of ligand and protein complementarity. We have placed this information in the Suplementary (Figure 1S).
  5. Of course, study of antiviral activity against SARS-CoV-2 is now very actually. Unfortunately, at the moment we are not able to conduct such research.

Round 2

Reviewer 2 Report

The paper has been improved and could be published now.